# Preventing the Distortion of CoO_6_ Octahedra of LiCoO_2_ at High-Voltage Operation of Lithium-Ion Battery: An Organic Surface Reinforcement

**DOI:** 10.3390/polym15092211

**Published:** 2023-05-06

**Authors:** Fu-Ming Wang

**Affiliations:** 1Graduate Institute of Applied Science and Technology, National Taiwan University of Science and Technology, Taipei 106335, Taiwan; mccabe@mail.ntust.edu.tw; Tel.: +886-2-2730-3755; Fax: +886-2-2737-6922; 2Sustainable Energy Center, National Taiwan University of Science and Technology, Taipei 106335, Taiwan; 3Department of Chemical Engineering, Chung Yuan Christian University, Taoyuan 320314, Taiwan; 4R&D Center for Membrane Technology, Chung Yuan Christian University, Taoyuan 320314, Taiwan

**Keywords:** organic, in situ, XAS, DSC, high-voltage, LCO

## Abstract

Lithium cobalt oxide (LiCoO_2_, LCO) has been widely used in electronic markets due to its high energy density and wide voltage range applications. Recently, high-voltage (HV, >4.5 V) operation has been required to obey the requirements of high energy density and cycle life in several applications such as electric vehicles and energy storage. However, the HV operation causes structure instability due to the over de-lithiation of LCO, as well as decomposing common carbonate solvents, thereby incurring the decay of battery performance. Moreover, a distortion of the CoO_6_ octahedra of LCO during de-lithiation induces a rehybridization of the Co 3d and O 2p orbitals. According to above reasons, decreasing the Co-O covalent bond promptly triggers high risks that significantly limit further use of LCO. In this research, an organic surface reinforcement by using bismaleimide–uracil (BU) that electrochemically forms a cathode electrolyte interphase (CEI) on LCO was explored. The results of electrochemical impedance spectroscopy and battery performance, such as the c-rate and cyclability tests, demonstrated that the modified CEI formed from BU significantly prevents the distortion of CoO_6_ octahedra. X-ray photoelectronic spectroscopy and in situ XAS indicated less LiF formation and higher bond energy of Co-O improved. Finally, the differential scanning calorimetry showed the onset temperature of decomposition of LCO was extended from 245 to 270 °C at 100% state of charge, which is about a 25 °C extension. The exothermic heat of LCO decreased by approximately 30% for high-safety use. This research confirms that the BU is eligible for high voltage (>4.5 V) LCO and presents outstanding electrochemical properties and safety performances.

## 1. Introduction

The lithium-ion battery (LIB) has been developed for more than 30 years since being commercialized by Sony in 1991. The first successful and most popular cathode material is LiCoO_2_ (LCO); created by Prof. Goodenough in 1980 and which led to his Nobel Prize in Chemistry award in 2019. With the explosion of the market in products such as portable electronics, electric vehicles (EVs), and energy storage, and the effects of COVID-19, the requirement for LIBs has increased dramatically in the past three years. The use of Ni-rich three-element-layered compounds and LiFePO_4_ (LFP) compounds are gradually instead of LCO owing to the limited natural availability of Co. However, Ni-rich layered cathodes suffer serious problems with cation mixing and safety issues, as well as many difficulties in manufacturing; thus, the cycle life of Ni-rich layered cathodes is not satisfactory for long-term usage and requires carefully controlled moisture content during processing. For LFP, the material price is cheaper than the Ni-rich layered compounds, but the weight is heavy and dramatically decreases its energy density to reach the same value as layered compounds. In addition, the dissolution of transitional metal ions, such as Ni^2+^ and Fe^2+^, also plays a critical role in the decay of battery performance, typically in high-temperature applications [1,2,3].

By considering with balance between cost and performance, LCO still demonstrates its outstanding characteristics from the point of view of its intrinsic material and further applications of the battery. Unfortunately, the capacity of LCO is not high enough due to extraction amounts of lithium ions in its layered structure, with only 50–55% of its original (140–150 mAh g^−1^), which is much lower than the Ni-rich layered (180–220 mAh g^−1^) and LFP (170–180 mAh g^−1^). Although the loading density of LCO can be designed up to 4.0 mg cm^−2^ and used to increase the area capacity, this represents more LCO being needed to maintain the energy density. To solve this problem, several studies have been undertaken, such as increasing the working voltage to more than 4.5 V by doping Al^3+^ [4], organic grafting [5], coating metal oxides by atomic layer deposition [6], and adding electrolyte additive [7]. In fact, some of those modified materials have already been commercialized and adopted for the production of batteries, but only with high-voltage maintenance without any guarantee of safety.

In this work, an organic reinforcement using bismaleimide–uracil (BU) as an oligomer has been investigated, to form a cathode electrolyte interphase (CEI) on the surface of LCO. BU is found to improve the distortion of the CoO_6_ octahedra of LCO during repeatable de-lithiation, thereby reducing the rehybridization of the Co 3d and O 2p orbitals, as well as enhancing the Co-O bond covalently. According to the results of this work, an interaction between BU and CoO_6_ of the LCO significantly promotes an excellent correlation between further battery performance and safety.

## 2. Experiment

### 2.1. Synthesis of BU 

Bismaleimide (Acros Organics, Geel, Belgium, 97%) and uracil (Acros Organics, 99%) were mixed in N-methyl-2-pyrrolidinone (NMP, Acros Organics, 99%) through continual stirring in an oil bath for 30 min at 130 °C. The total solid content of the resultant BU solution was optimized to 7 wt%. The resulting solution was stored in a refrigerator to prevent thermal and self-polymerization before reaction on the LCO particle surface. Figure 1 shows the structure of the BU oligomer and the expected representation of BU on the LCO surface.

### 2.2. Electrode Preparation

In this study, we used LCO as the cathode material, Li foil as the reference electrode, a polypropylene microporous film (Celgard 2320) as a separator, Super-P (SP) (Timcal) as the conductive agent, and polyvinylidene difluoride (PVDF) as a binder. All chemicals were of reagent grade and used as received in a glove box. The LCO electrode was fabricated by preparing a slurry of active material: 90 wt% LCO, 5 wt% SP, and 5 wt% PVDF in a solvent of NMP. The prepared slurry was coated onto Al foil and dried at 110 °C for 2 h. The dried electrodes were then pressed uniformly and punched into 12 mm diameter disks for the half-cell test. The BU was added to prepare the slurry with a solid content of 1 wt% based on the amount of LCO. The reaction process of the BU with LCO in the slurry was at room temperature for 3 h. This reaction guarantees the surface of the LCO was homogeneously covered with BU. The average area capacity of the two LCO electrodes was controlled at 3.5 mAh cm^−2^.

### 2.3. Electrolyte Preparation

An electrolyte was prepared for the experiments with 1 M lithium hexafluorophosphate (LiPF_6_) in ethylene carbonate (EC) and ethyl methyl carbonate (EMC) (volume ratio of 1:2, battery grade, water content less than 20 ppm), purchased from Uni-onward Inc, Taipei, Taiwan. The battery fabrication and electrolyte preparation were performed inside a glove box in an Ar gas atmosphere to avoid the influence of moisture.

### 2.4. Study of Electrochemical Performance

To investigate the electrochemical behavior of the cells, a CR2032-type coin cell was employed. Cyclic voltammetry (CV) (VMP3, BioLogic Science Instruments, Glossop, UK) plots of the cells versus Li/Li^+^ were obtained at a scan rate of 0.01 mV/s in the potential range of 3–5.0 V. Charge–discharge was measured using a battery testing instrument (BAT-750B Battery Automatic Tester) at a constant current of 0.1 C at 2.8–4.5 V versus Li/Li^+^ and constant voltage charges at 4.5 V when the current was lower than 0.01 C in the charging process. The discharging process was controlled at a constant current of 0.1 C at 4.5–2.8 V versus Li/Li^+^. Electrochemical impedance spectroscopy (EIS) from 100.0 K to 0.1 Hz with an AC amplitude of 5 mV was adopted by using VMP3 and an EC-Lab electrochemical software, produced by Biologic Inc., Seyssinet-Pariset, France for further model simulation.

### 2.5. Effect of Electronic Configuration on Synchrotron Radiation

In situ electronic structure analysis of the electrodes (after disassembly in the glove box) was performed using synchrotron hard X-ray absorption spectroscopy (XAS). The hard XAS was performed in beamlines BL01C1 and 17C1 at the National Synchrotron Radiation Research Center (NSRRC), Hsinchu, Taiwan. The hard XAS beamline covered the spectral range of 5–40 keV, with an average energy-resolving power of 5 keV. The soft XAS beamline covered the spectral range of 60–1250 eV, also with an average energy-resolving power of 60 eV. The end station was primarily designed for studying condensed—I have modified, please check. phase photochemistry and the electronic structures of new materials. In situ experiments were performed at C/10 in a modified coin cell. A Si (111) monochromatic double crystal was used to perform the energy scan, in which the parallelism was adjusted to eliminate high-order harmonics. All spectra were obtained in transmission mode. Ionization chambers were used as detectors to monitor the intensity of the incident beams on and transmitted beams through the specimen, enabling the calculation of the absorption coefficient from the logarithm of the intensity ratio of the incident and transmitted beams.

### 2.6. Morphology and Exothermic Heat Analysis

A scanning electron microscopy (SEM) was used to observe the morphology of electrodes (after Pt coating the electrode in a glove box) at an accelerating voltage of 15 kV, using a LEO-1530 microscope. The samples were placed in a custom-built high-vacuum stainless-steel holder to transfer the electrodes from the dry room to the SEM instrument. The electrodes were not influenced by the treatment before assembling and drying the cell. 

Once the cells were fully charged (100% SOC to 4.5 V), they were disassembled in a glove box, washed with pure EMC solvent, and dried in a vacuum oven for 1 h. The dried and recovered electrodes were soaked with a suitable amount of fresh electrolyte. Approximately 4 ± 0.1 mg of sample was measured for thermal analysis by using a differential scanning calorimeter (DSC). The measurement for the pristine and modified cathode electrodes was performed in the temperature range from room temperature to 350 °C at a heating rate of 5 °C min^−1^, under a nitrogen flow rate of 50 mL min^−1^. X-ray photoelectron spectroscopy (XPS) (PHI, 1600S) was adopted to analyze surface composition.

## 3. Results and Discussion

Appendix A shows the IR spectra, displaying several characteristic absorptions at different wavenumbers, such as two C=O stretches from the uracil structure at 1748 and 1670 cm^−1^. In addition, the stretching C=C vibration at 759 cm^−1^, as well as two N-H stretching at 1374 and 596 cm^−1^, appeared from the uracil structure. Furthermore, the stretching C=C vibration at 1516 cm^−1^ from the bismaleimide structure was also found. After finishing the synthesis of the BU oligomer, the C=C bonds of uracil and bismaleimide had almost disappeared, indicating the synthesis reaction was completed. In the power view of the electrodes, several characteristic absorptions of BU were found on the modified LCO electrode, although the peak intensity was low due to the tiny amount of organic layer that was covered. On the contrary, pristine LCO did not reveal any characteristic absorptions, confirming that BU was on the surface of LCO. However, from the XRD results, shown in Appendix A, the modified LCO electrode that contained BU did not display much difference because this organic coverage did not reveal any crystallinity.

Figure 1a shows the charge–discharge profile of the pristine and modified LCO electrodes. According to the results, the pristine LCO electrode delivered an initial discharge capacity of 180.2 mAh g^−1^ with a coulombic efficiency (CE) of 91.1%; this is a common capacity and CE of LCO when operated in a high voltage window (4.5–2.8 V). The modified LCO electrode with BU additive delivered an initial discharge capacity of 179.6 mAh g^−1^ with a CE of 89.4%. The CE of the modified electrode was a little lower than that of the pristine one, indicating that BU causes polarization initially owing to the electrochemical reaction of BU. In fact, the charge curve increased faster than the pristine electrode in a range from 3.9 to 4.2 V; this affection can be concluded to be due to the electrochemical reaction of BU. Typically, this is the reaction voltage range of Co^3+^ oxidized to Co^4+^—the main reaction of LCO. Figure 1b shows the rate performance of the two LCO electrodes. From the results, the difference between the two LCO electrodes was observed from 0.2 C. The modified electrode displayed a higher discharge capacity than that of the pristine electrode when the current rate increased to 1 C. Even when the current rate gradually returned to 0.1 C, the pristine electrode, however, cannot recover back to its original capacity. Normally, the capacity of the LCO will be returned easily when charged at only 1 C rate (relatively, not too high) because the LCO has a greater electrical conductivity than other cathode materials such as LFP. However, the LCO was charged to 4.5 V in this rate test, presenting the de-lithiation percentage of LCO as more than 50% and causing the distortion of the CoO_6_ octahedra, thereby increasing the rehybridization of the Co 3d and O 2p orbitals, as well as reducing the Co-O bond covalently [8,9]. In fact, LCO suffering the distortion of CoO_6_ will lead to O_2_ evolution and a high risk of explosion. On the contrary, the modified LCO electrode provides greater rate performance compared with the pristine, indicating the BU stabilizes the LCO structure at a high voltage range operation. Figure 1c shows the cycle performance at room and high (55 °C) temperatures of the two LCO electrodes. For the first 10 cycles, the pristine electrode decays fast at room temperature due to the distortion of CoO_6_. After 10 cycles, the testing temperature was increased to 55 °C, thereby increasing the capacity, owing to the increment of ionic conductivity of the electrolyte. However, the cycle stability does not present as being stable, with thewobbly curve behavior of the pristine electrode. Although further high-rate performance (0.5 C) results show the aforementioned behavior becomes stable, the capacity presents less than 100 mAh g^−1^, with a poor retention of 55.3% at significantly higher temperatures and high voltage. In the same testing protocol, the modified electrode delivered 133.7 mAh g^−1^, with a better retention of 74.7% compared with the pristine. To verify the constant rate (0.2 C/0.2 C) in the cycle test, Figure 1d shows two electrodes were operated at room temperature for 50 cycles. According to the results, the modified electrode containing BU displayed better performance and had a higher capacity and cycle retention (153.9 mAh g^−1^; 85.9%) than the pristine (133.1 mAh g^−1^; 73.6%).

Figure 2 shows the EIS spectra of the two LCO electrodes at different cycles, the 1st and the 60th cycles of Figure 1c. After the formation of the two electrodes, Figure 2a clearly demonstrates the two LCO electrodes have two semi-circles, presenting the impedance of CEI (R_cei_) and the charge transfer resistance (R_ct_) of LCO surface. The R_cei_ of the pristine and modified electrodes are 50.7 and 89.3 Ω, and the R_ct_ of the pristine and modified electrodes are 685.1 and 896.2 Ω, respectively. Because there are only two semi-circles in the EIS spectra in comparison with two electrodes, this indicates the BU had joined the CEI formation and increased the interface impedance of the BU characteristics in the first cycle. It can be speculated that the CEI formed from the BU and carbonate electrolyte is different from the traditional one from that of carbonate electrolyte only. Interestingly, the Warburg impedance also demonstrates typical diffusion behavior in terms of the interface to the bulk LCO material. The Warburg impedance, in fact, is not only used to calculate the value of diffusion resistance but the diffusion can also be observed through the distinguishment of the angle, by figuring out if it is controlled by resistance or capacitance [10]. According to the result in Figure 2a, the angle of Warburg impedance of the modified electrode is 35.7°, which is lower than that of the pristine electrode (40.2°). Previous literature has discussed that an angle of Warburg impedance lower than 45° presents that the diffusion is controlled by resistance, and capacitance is controlled at higher than 45°. With an angle of 45°, it demonstrates a perfect diffusion without any resistance and capacitance effects. By adding the BU, the diffusion of LCO obviously affects the resistance and dramatically impedes the electrochemical reaction; thus, the R_ct_ has a higher value. Because the diffusion of the modified electrode is controlled by resistance, it indicates that the CEI forms from BU and is not used to store electrons in its chemical composition. Figure 2b shows there are still two semi-circles after cycling in Figure 1c, indicating that electrochemical cycling does not form other compounds and interfaces. However, the R_cei_ and R_ct_ of the pristine and modified electrodes was changed by the effects of cycling. The R_cei_ were 79.4 and 50.8 Ω, and the R_ct_ of the pristine and modified electrodes were 858.4 and 493.2 Ω, respectively. These results demonstrate that the pristine LCO was charged to 4.5 V and caused possible side reactions of O_2_ evolution and electrolyte decomposition from the distortion of CoO_6_ octahedra. Thus, the CEI has a high impedance due to a thick layer formation or dense structure. Accompanying the high R_cei_, the electrochemical reaction will not easily perform on the interface with a high R_ct_ value, which was almost 1.5 times more than the modified electrode. The BU indeed improved those aforementioned problems of LCO charged to high voltage with cycling. Moreover, the angle of Warburg impedance of the modified electrode was then 30.1°, which was still lower than that of the pristine electrode (38.6°) owing to the BU effects.

Figure 3 shows the CV results of the two LCO electrodes, which was used to analyze the valence changes and kinetic behavior of the Co ions. Figure 3a shows that there were two redox couples of the pristine LCO in the CV result. The first redox couple was the main reaction of Co^3+^ to Co^4+^ at 4.2 V and Co^4+^ to Co^3+^ at 3.8 V, respectively [8]. The second redox couple was the Li ion arrangement in the CoO_2_ framework of ordering at 4.6 V and disordering at 4.3 V [8]. With the BU addition, the modified electrode displayed a delayed reaction of Co^3+^ to Co^4+^ at 4.45 V but no certain effect on the reaction of Co^4+^ to Co^3+^. The polarization of the modified electrode only took place at the first anodic reaction, indicating that the BU needs an activation for its CEI formation on the LCO surface but had no issue after the CEI had been competely formed in the cathodic scan. From the point of view of this CEI formation, the figure inside Figure 3a displays a reaction took place at around 3.45 V in the modified electrode, but the pristine electrode does not have this reaction peak. This observation explains that the BU is electrochemically oxidized before the reaction Co^3+^ to Co^4+^, the reaction Co^3+^ to Co^4+^ is, therefore, delayed to 4.6 V, but other reactions, including the ordering of Li ion arrangement in the CoO_2_ framework and two cathodic reactions, showed almost no difference. To verify the BU effects on the second scan of the modified electrodes, the experiment of Figure 3b is adopted. In Figure 3b, all the CV curves are stacked together with no difference in behavior. In addition, the BU oxidization no longer exists in the second scan. Those results confirm that the BU only reacted in the first scan, and with no polarization once the CEI was completely formed. These two CV results are abided by the explanations of charge and discharge profiles in Figure 2.

Figure 4 shows the SEM images of the two LCO electrodes after cycling in Figure 1c. Figure 4a displays a single crystal particle behavior of LCO with a size of approximately 30–80 μm. The LCO particles are surrounded by SP (100 nm). Figure 4b has zoomed in on the particle surface, which is clear and smooth, with the traditional CEI formation. With only 60 cycles in Figure 1c, LCO particle crack and structure collapse were not found, like in the literature [11,12]. Figure 4c shows the modified particles with the BU additive; the shape of LCO does not reveal much difference compared with the pristine. However, Figure 4d displays that the BU has affected the surface and presents a uniform distribution but roughness type of morphology of CEI coverage on the LCO surface.

It is interesting to understand the chemical composition of CEI formed from BU, and its contribution to battery performance enhancement. Figure 5 shows the XPS analysis of the two LCO electrodes after cycling in Figure 1c. Figure 5a shows the O1s spectra composed of Li_2_CO_3_ (~532.0 eV) and C=O related organic compounds on the pristine LCO surface after cycling, indicating that the CEI was covered by the electrolyte decomposition [13]. In fact, the CEIs formed on the LCO surface are mostly Li_2_CO_3_, LiF, and C-H/C-O- related substances, typically from the decomposition of EC [14,15]. For the modified LCO electrode, the BU covered on the LCO surface is used to prevent electrolyte decomposition; it is, thus, the formation of Li_2_CO_3_ that is significantly decreased. A similar result is shown in the F1s spectra in Figure 5b. The composition of CEI on the pristine LCO surface not only contains Li_2_CO_3_, but the LiF (~685.5 eV) is also dramatically increased after cycling. Previous studies have discussed the formation of LiF mainly comes from the LiPF_6_ decomposition; this high impedance substance will inhibit the ionic transfer and leads to a high ohmic polarization [13,14]. This result explains that a perfect long cycling performance needs to overcome CEI formation of the electrode surface. On the contrary, Figure 5b clearly shows the formation of LiF is indeed prevented by the BU coverage as well as the PVdF (~688.0 eV). This result demonstrates the modified electrode was prepared by LCO covered with BU, and PVdF does not react BU, indicating that the binder intrinsic and electrode mechanical properties are maintained. The aforementioned XPS analysis has proven that the BU inhibits electrolyte decomposition on the LCO surface and is highly stable during electrode cycling. Appendix A shows the pre-cycle and post-cycle XRD of pristine and modified LCO electrodes. In terms of the results, the pristine and modified electrodes do not show much difference due to the tiny amount of BU that covers the LCO surface. Interestingly, Figure 1c shows the (003) facet of pristine LCO significantly decreases after cycling, which may be used to correlate to the thick SEI that was formed owing to the electrolyte decomposition. However, the modified LCO electrode still maintains its intensity by BU coverage. This result is compatible with the XPS analysis in Figure 5, as well as the battery performance measurements.

To verify the changes in bulk properties, Co ion valence, and the bond length of the LCO by BU coverage, hard XAS analysis was adopted. Figure 6a,b show the K-edge X-ray absorption near edge spectroscopy (XANES) of the pristine and modified electrodes in terms of different states of charge (SOC) in the first charge. According to the literature, the LCO of this work belongs to O3-type material [5]. Figure 6a demonstrates that the pristine LCO electrode has a pre-edge peak around 7710 eV. This peak corresponds to the transition of the 1s electron to an unoccupied 3d orbital of the Co^3+^ ion with a low-spin electronic configuration. Several studies have concluded that an appearance and symmetry pre-edge peak correlates to pure electric quadrupole coupling and an ideal CoO_6_ octahedral [16]. By increasing the SOC, this pre-edge peak was shifting to high energy, indicating the noncentrosymmetric environment of distorted CoO_6_ octahedral was gradually formed [9,17]. The peaks around 7718 and 7728 eV are assigned to the two states of the LCO. Some research has demonstrated that these two states are correlated to the shakedown process of the ligand-to-metal transfer, as well as without the shakedown process. From the point of view of the disappearance of the lower energy peak (~7718 eV) during charging, the lithium ions de-intercalated, which leads to a local structural distortion around the Co atom. Another explanation is concluded to the mismatched Co 3d–O 2p orbital overlap. In Figure 6a, the pristine electrode shows the weakening of this peak started from 0% SOC, and mostly disappeared at 50% SOC, indicating the local structural distortion was triggered at the beginning of de-intercalation. Moreover, the valence of the Co ion was changed to high valance according to the peak shifting from 7727 eV to 7732 eV, indicating the formation of Co^4+^. However, Figure 6b displays that those three peaks (7710, 7718, and 7728 eV) of the modified electrode were maintained as stable from 0 to 25% SOC, indicating the BU is used to delay the de-intercalation as well as adjusting local structural distortion of LCO. In fact, the modified LCO showed a much clearer difference from 25 to 50% SOC and the peaks at 7718 and 7728 eV displayed a better CoO_6_ octahedral structure. By adding the BU, the Co ion valence of the modified electrode at 50% SOC showed lower energy than the pristine electrode, suggesting the working reaction of BU significantly affected the LCO within the range of 0 and 50% SOC. This result is compatible with Figure 1a and Figure 3a, where a charging or anodic polarization curve was found from 3.95 V, at approximately 50% capacity of the LCO. Figure 6c,d shows the extended X-ray absorption fine structure spectroscopy (EXAFS) of the pristine and modified electrodes. There were two major peaks in the *k*^3^-weighed Fourier transform spectra at the Co K-edge that corresponds to Co-O and Co-Co bonds [15]. Both the magnitudes of the two major bonds were changed at different SOCs due to the local structure distortion of CoO_6_. Figure 6c shows the bond energy significantly decreased by increasing the SOC, indicating the Co ions are under a higher oxidative environment of excited 2p electrons in the pristine electrode. Interestingly, it was also found that the Co-O bond energy decreased earlier than the Co-Co bond at 25% SOC, which demonstrates that the first change in the effective nuclear charge of Co ions was correlated with the O atoms. On the contrary, Figure 6d shows that the Co-O bond energy was maintained strongly when charged to 25% SOC in the modified electrode. This result suggests that the BU collaborates with the Co-O bond rather than the Co-Co bond, and further stabilizes the CoO_6_ octahedral structure during cycling.

Figure 7 shows the DSC analysis of the exothermic behavior of the two electrodes at 100% SOC. The pristine electrode starts its thermal reaction from 168 °C and further induces two huge exothermic reactions at 245 and 261 °C. This is because the highly de-lithiated LCO had less stability on the Co-O and Co-Co bonds, suggesting the gas evolved from structure collapse [5], typically O_2_ generation. The heat release of the de-lithiated pristine electrode was calculated to be 541.9 J g^−1^. With the BU addition, the de-lithiated modified electrode extended its first thermal reaction to 221 °C, a nearly 50 °C delay. In addition, the original two huge exothermic reactions in the pristine electrode were changed to only one major reaction in the modified electrode; this reaction was at approximately 270 °C. Compared with the pristine electrode, the onset temperature of the major reaction was dramatically postponed by 25 °C. In the meantime, the exothermic heat release of the modified electrodes was calculated to be 381.1 J g^−1^, which was about a 29.7% decrease compared with the pristine. This result confirms the XANES and EXAFS analysis that the Co-O bond is enhanced by the BU in the range of 0–50% SOC and, thus, inhibits O_2_ release for further thermal reaction.

## 4. Conclusions

This work demonstrates an organic surface reinforcement of CEI formation on the LCO surface by a BU additive. Within the high voltage window operation, the CEI formed from BU significantly mitigates the distortion of CoO_6_ and stables Co-O bond covalently, typically in the range of 0–25% SOC (initial charge range). By an investigation of this research, BU is suggested as having a collaboration with the Co-O bond rather than Co-Co bond and, thus, dramatically extends the onset temperature and exothermic heat release. BU is a new electrode additive for high safety in high-voltage LCO material and has a high potential in further EVs applications. 

## Data Availability

Data will be provide on request.

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
