# Peer review of "Preventing the Distortion of CoO6 Octahedra of LiCoO2 at High-Voltage Operation of Lithium-Ion Battery: An Organic Surface Reinforcement"

_polymers, 2023, doi:10.3390/polym15092211_

Round 1

Reviewer 1 Report

In this manuscript, the author reported an organic surface reinforcement method by using bismaleimide-uracil (BU) that electrochemically forms a cathode electrolyte interphase (CEI) on LCO. the modified CEI forms from BU significantly prevents the distortion of CoO6 octahedra and improves the battery performance of LCO at high-voltage. This method provides a new idea for the development of LCO at high-voltage operation of lithium-ion battery, but there are some issues that need to be addressed.

1. In the abstract, the description of “decreasing the Co-O bond covalently” is wrong, should be changed to “decreasing the Co-O covalent bond”.

2. This manuscript mentions an organic surface reinforcement by using BU that electrochemically forms a cathode electrolyte interphase (CEI) on LCO, but the experiment part only provides the synthesis process of BU, and does not describe the process of introducing BU to the LCO surface.

3. The characterization of the electrode material before cycling, including XRD, SEM, XPS, FT-IR, Raman, etc., is not provided in the paper, so it is impossible to determine whether the BU is successfully introduced into the LCO surface.

4. In the result and discussion part, the article only provides the rate capabilities test at room temperature and 55°C, and should be supplemented with cycling performance at different current densities to confirm its cycle life and cycle stability.

5. The XPS in Figure 5 was not fitted to the peaks and it is not possible to determine whether the bonding described by the authors exists.

6. Supplementary pre-cycle and post-cycle XRD is recommended to verify the changes in the crystal structure of LCO. 

Author Response

Dear Editor,

Thank you for your letter regarding the reviewer’s comments. These comments suggest modifications. In the attached “revision letter” copy, each change has been distinguished clearly from the earlier version of our manuscript. In addition to the specific modifications to the text, general explanations in response to their comments are listed as follows:

Reviewer 1:

In this manuscript, the author reported an organic surface reinforcement method by using bismaleimide-uracil (BU) that electrochemically forms a cathode electrolyte interphase (CEI) on LCO. the modified CEI forms from BU significantly prevents the distortion of CoO6 octahedra and improves the battery performance of LCO at high-voltage. This method provides a new idea for the development of LCO at high-voltage operation of lithium-ion battery, but there are some issues that need to be addressed.

Ans: Thanks for reviewer’s comment. We have provided a revised letter concerning those comments and suggestions from reviewers.

  1. In the abstract, the description of “decreasing the Co-O bond covalently” is wrong, should be changed to “decreasing the Co-O covalent bond”.

Ans: Thanks for reviewer’s comment. We have modified abstract in accordance with this suggestion.

  1. This manuscript mentions an organic surface reinforcement by using BU that electrochemically forms a cathode electrolyte interphase (CEI) on LCO, but the experiment part only provides the synthesis process of BU, and does not describe the process of introducing BU to the LCO surface. 

Ans: Thanks for reviewer’s comment. We have added a description of reaction process of BU with LCO slurry in page 4, line 20-23. A detail process is mentioned, “the BU is added to prepare slurry with a solid content of 1 wt% based on the amount of LCO. The reaction process of BU with LCO in slurry is at room temperature for 3 hours. This reaction guarantees the surface of LCO is homogeneously covered with BU”.

  1. The characterization of the electrode material before cycling, including XRD, SEM, XPS, FT-IR, Raman, etc., is not provided in the paper, so it is impossible to determine whether the BU is successfully introduced into the LCO surface.

Ans: Thanks for reviewer’s comment. We have performed XRD and FTIR measurements of pristine and modified LCO electrodes as well as BU oligomer and urial precursor. According to the results in Fig. S1a, the IR spectra shows several characteristic absorptions at different wavenumbers such as two C=O stretches from uracil structure at 1748 and 1670 cm-1. In addition, the stretching C=C vibration at 759 cm-1 as well as two N-H stretching at 1374 and 596 cm-1 are appeared from uracil structure. Furthermore, the stretching C=C vibration at 1516 cm-1 from bismaleimide structure is also found. After finishing the synthesis to BU oligomer, the C=C bonds of uracil and bismaleimide are almost disappeared, indicating the synthesis reaction is completed. In the power view of electrodes, several characteristic absorptions of BU are found on modified LCO electrode, although the peak intensity is low due to tiny amount of organic layer is covered. On the contrary, pristine LCO cannot reveal any characteristic absorptions, this result confirms BU is on the surface of LCO.

However, from XRD results in Fig. S1b, the modified LCO electrode that contains BU does not display much different, because this organic coverage will not reveal any crystallinity. We have added this description in page 6, line 13-23.

  1. In the result and discussion part, the article only provides the rate capabilities test at room temperature and 55°C, and should be supplemented with cycling performance at different current densities to confirm its cycle life and cycle stability.

Ans: Thanks for reviewer’s comment. We have performed the cycle tests under a constant rate. According to our results in Fig. 1d, the BU modified LCO provides excellent cycle performance. The modified electrode containing BU displays better performance that has higher capacity and cycle retention (153.9 mAh g-1; 85.9%) than pristine (133.1 mAh g-1; 73.6%). We have added this description in page 7, line 22-26.

  1. The XPS in Figure 5 was not fitted to the peaks and it is not possible to determine whether the bonding described by the authors exists. 

Ans: Thanks for reviewer’s comment. We have fitted XPS spectra to verify the description of each peaks in Fig.5.

  1. Supplementary pre-cycle and post-cycle XRD is recommended to verify the changes in the crystal structure of LCO. 

Ans: Thanks for reviewer’s comment. Fig. S1 shows the pre-cycle and post-cycle XRD of pristine and modified LCO electrodes. In terms of the results, the pristine and modified electrodes do not show much different due to tiny amount of BU is covered on LCO surface. Interestingly, Fig. 1c shows the (003) facet of pristine LCO significantly decreases after cycling, which may use to correlate to the thick SEI was formed owing to the electrolyte decomposition. However, the modified LCO electrode still maintain its intensity by BU coverage. This result is compatible to XPS analysis in Fig. 5 as well as battery performance measurements. We have added this description in page 10, line 11-17.

Reviewer 2 Report

The author utilized the polymer of bismaleimide-uracil to improve the high-voltage performance of LiCoO2 cathode and studied the enhanced mechanism carefully. Based on the abundant experience, it can be seen that the electrochemistry is solid and the logic is smooth, and I really appreciate the research attitude. As a result, I recommend a minor revision of this paper, and some questions are listed below.

1.      The elevated performance is ascribed to the reduced rehybridization of Co and O orbitals, as well as the enhanced Co-O covalent bond. Despite the latter has been proved by XANES and EXAFS, the former (maybe theoretical?) lacks practical evidence to confirm. What is the relationship between the molecular structure of added polymer (such as BU) and the composition of CEI on the LCO cathode?

2.      As shown in Figure 1b-c, the capacity degradation is not mitigated actually, or even very drastic in 55 oC, which indicates the improvement is limited. 1) Are there any better strategies, such as infusing BU into bulk LCO? 2) Importantly, why the thin surface coating of BU can largely change the behavior of bulk LCO and thick electrodes (3.5 mAh cm-2)?

3.      No obvious cracks and collapse are found in the SEM with a scale bar of 100 nm, how about the higher magnification or TEM images?

4.      It is no doubt that the thermal stability of LCO is improved with BU, so the 100% SOC LCO is tested. In practice, the temperature changes as the structure evolve at the same time, other than maintaining 100% SOC along the thermal cycling. Are there any better solutions to test the thermal stability upon cycling in an actual working condition, such as in-situ or real-time tests?

Author Response

Reviewer 2:

The author utilized the polymer of bismaleimide-uracil to improve the high-voltage performance of LiCoO2 cathode and studied the enhanced mechanism carefully. Based on the abundant experience, it can be seen that the electrochemistry is solid and the logic is smooth, and I really appreciate the research attitude. As a result, I recommend a minor revision of this paper, and some questions are listed below.

Ans: Thanks for reviewer’s comment. We are doing our best in responding those comments from reviewer.

  1. The elevated performance is ascribed to the reduced rehybridization of Co and O orbitals, as well as the enhanced Co-O covalent bond. Despite the latter has been proved by XANES and EXAFS, the former (maybe theoretical?) lacks practical evidence to confirm. What is the relationship between the molecular structure of added polymer (such as BU) and the composition of CEI on the LCO cathode?

Ans: Thanks for reviewer’s comment. This is a good question concerning the reaction of BU to LCO surface. We expect that the oxygen atom on C=O of BU may use to collaborate with Co-O of LCO rather than Co-Co. This strong electron withdrawing group is thinking to have electron attraction and interaction from Co-O and therefore provide a better CoO6octahedral structure by BU addition. However, more analysis and measurements are needed to further discuss.

  1. As shown in Figure 1b-c, the capacity degradation is not mitigated actually, or even very drastic in 55 oC, which indicates the improvement is limited. 1) Are there any better strategies, such as infusing BU into bulk LCO? 2) Importantly, why the thin surface coating of BU can largely change the behavior of bulk LCO and thick electrodes (3.5 mAh cm-2)?

Ans: Thanks for reviewer’s comment. We have added one new figure (Fig. 1d) to illustrate the cycle performance tested at a contact rate (0.2C/ 0.2C) and at room performance. According to all results shown in Fig. 1, including rate, cycled at 55oC, and cycled at room temperature are displayed better than the pristine, indicating the BU improves LCO electrochemical performance.

Although the XANES and EXAFS analysis already confirm that the Co-O bond is enhanced by the BU in the range of 0-50% SOC and thus inhibits O2 release for further thermal reaction. Typically, the CEI forms from BU significantly mitigates the distortion of CoO6 and stables Co-O bond covalently. In fact, we are still thinking why tiny amount of BU is able to improve LCO electrochemical performance. We expect that the oxygen atom on C=O of BU may use to collaborate with Co-O of LCO rather than Co-Co. This strong electron withdrawing group is thinking to have electron attraction and interaction from Co-O and therefore provide a better CoO6 octahedral structure by BU addition. However, more analysis and measurements are needed to further discuss.

  1. No obvious cracks and collapse are found in the SEM with a scale bar of 100 nm, how about the higher magnification or TEM images?

Ans: Thanks for reviewer’s comment. We have measured TEM for two electrodes after cycling. Unfortunately, there are not much different between each other, the reason may cause by the cycle test is not long as possible. Therefore, only SEM reveals the CEI different on surface morphology.

  1. It is no doubt that the thermal stability of LCO is improved with BU, so the 100% SOC LCO is tested. In practice, the temperature changes as the structure evolve at the same time, other than maintaining 100% SOC along the thermal cycling. Are there any better solutions to test the thermal stability upon cycling in an actual working condition, such as in-situ or real-time tests?

Ans: Thanks for reviewer’s comment. Normally, operando GCMS or adiabatic rate calorimeter (ARC) may use to determine gas evolution and calorimetric from a battery system. In our case, the O2 will release from LCO layered structure at high temperature and high SOC. The operando GCMS and ARC are able to deliver important information at such conditions aforementioned.

Round 2

Reviewer 1 Report

I suggest the acceptance of the revised manuscript.